# Current Approaches for Monitoring of Patients with Inflammatory Bowel Diseases: A Narrative Review

**DOI:** 10.3390/jcm13041008

**Published:** 2024-02-09

**Authors:** Alessandro Vitello, Marcello Maida, Endrit Shahini, Fabio Salvatore Macaluso, Ambrogio Orlando, Mauro Grova, Daryl Ramai, Gaetano Serviddio, Antonio Facciorusso

**Affiliations:** 1Gastroenterology and Endoscopy Unit, S. Elia Hospital, 93100 Caltanissetta, Italy; a.vitello@asp.cl.it (A.V.); marcello.maida@hotmail.it (M.M.); 2Department of Medicine and Surgery, School of Medicine and Surgery, University of Enna ‘Kore’, 94100 Enna, Italy; 3Gastroenterology Unit, National Institute of Gastroenterology-IRCCS “Saverio de Bellis”, 70013 Castellana Grotte, Italy; endrit.shahini@irccsdebellis.it; 4Inflammatory Bowel Disease Unit, Department of Medicine, A.O.O.R. “Villa Sofia-Cervello” Hospital, 90146 Palermo, Italy; fsmacaluso@gmail.com (F.S.M.); ambrogiorlando@gmail.com (A.O.); mauro.grova@gmail.com (M.G.); 5Division of Gastroenterology and Hepatology, University of Utah Health, Salt Lake City, UT 84132, USA; daryl.ramai@hsc.utah.edu; 6Department of Medical and Surgical Sciences, University of Foggia, 71122 Foggia, Italy; gaetano.serviddio@unifg.it

**Keywords:** monitoring, surveillance, biomarkers, IBD, ulcerative colitis, Crohn’s disease, endoscopy, intestinal ultrasound, video-capsule endoscopy

## Abstract

Background: Patients with inflammatory bowel diseases (IBD) require proactive monitoring both during the active phase to evaluate therapeutic response and during the remission phase to evaluate relapse or colorectal cancer surveillance. However, monitoring may vary between patients with ulcerative colitis (UC) and Crohn’s disease (CD), with distinct tools and intervals. Methods: This narrative review aims to focus on modern approaches to IBD monitoring, considering international guidelines and expert consensus. Results: The most recent European diagnostic guidelines advocate a combination of clinical, laboratory, endoscopic, and radiological parameters to evaluate the disease course of patients with IBD. Unfortunately, the conventional symptom-based therapeutic approach does not improve long-term outcomes and there is no single ideal biomarker available. Endoscopy plays a key role in evaluating response to therapy as well as monitoring disease activity. Recently, bedside intestinal ultrasound (IUS) has gained increasing interest and diffusion as it appears to offer several advantages including the monitoring of therapeutic response. Conclusion: In light of growing clinical advances, we present a schematic evidence-based monitoring algorithm that can be easily applied in clinical practice which combines all major monitoring modalities, including noninvasive tools such as IUS and video-capsule endoscopy.

## 1. Introduction

Inflammatory bowel diseases (IBD) involve complex chronic conditions that lead to long-term and/or permanent damage if not treated effectively [1]. Ulcerative colitis (UC) and Crohn’s disease (CD) are the two main forms of IBD, both characterized by immune-mediated inflammation in the gastrointestinal tract [1]. It is crucial to receive timely and effective treatment to prevent the progression of these diseases and avoid irreparable structural and functional damage [2,3,4]. Despite the growing number of drugs available for treating IBD, their long-term effectiveness remains suboptimal. Approximately one-third of IBD patients do not respond to initial therapies, while another third experiences incomplete response or lose their response over time [5]. While the therapeutic arsenal for IBD is constantly expanding, there is still a need for more effective treatments to improve patient outcomes [5]. Proactive monitoring is essential for IBD patients, both during the active phase to evaluate therapeutic response, and during remission to evaluate relapse or colorectal cancer (CRC) surveillance [6,7,8,9,10]. However, monitoring procedures can vary between patients with UC and CD, with different tools and time points used. The most recent ECCO-ESGAR diagnostic guidelines advocate a combination of clinical, laboratory, endoscopic, and radiological parameters to assess the success of IBD treatment(s) (Figure 1) [11,12].

## 2. Relevant Sections

### 2.1. Clinical Parameters

The initial approach for a patient with IBD is primarily clinical, and there are various clinical scores available to conduct the initial evaluation of disease severity. For instance, the Harvey–Bradshaw Index (HBI) is used in CD, while the partial Mayo Score (pMAYO) is used in UC. There has been a recent shift towards evaluating patient-reported outcomes (PRO2) because patients’ perceptions of their illness and symptoms may differ significantly from those of the treating physician [7,13,14,15]. On the other hand, patients often consider clinical symptoms as the most crucial factor in addressing their care, and therefore can also represent important therapeutic goals [16]. However, about one-third of IBD patients in clinical remission will have transmural mucosal inflammation [13,17,18]. As a result, the STRIDE-II consensus recognized symptom relief (clinical response and clinical remission) as crucial short-term and intermediate therapeutic goals. However, clinical parameters alone are insufficient as long-term targets, and the use of objective markers is necessary [7]. It is worth noting that the CALM trial indicated that escalating treatment based on symptoms alone resulted in a lower rate of mucosal healing (MH) than guiding treatment with a composite strategy of clinical and biochemical activity evaluation, which includes fecal calprotectin (FC) and C-reactive protein (CRP) [19]. According to the latest American Gastroenterological Association (AGA) guidelines, patients with symptomatically active IBD should follow a combination approach that includes biomarkers in addition to symptoms [20,21]. This approach is recommended for better disease management.

### 2.2. Laboratory Parameters

A biomarker is defined as “a characteristic that is objectively measured and evaluated as an indication of normal biologic processes, pathogenic processes, or pharmacologic responses to a therapeutic intervention” [22]. An ideal biomarker should have three qualities. Firstly, it must be reliable in terms of sensitivity, specificity, and reproducibility. Secondly, it should have good patient compliance, which means it should be non-invasive, rapid, and cheap. Finally, it should have kinetic stability, which means it should not be influenced by the patient’s status and remain stable during storage and shipment [23]. Over the years, numerous biomarkers have been investigated throughout the years, and most of them are summarized in Table 1. They are divided into two main categories: serum biomarkers and fecal biomarkers [23]. According to a meta-analysis by Mosli et al., where the diagnostic performance of the main biomarkers was compared with endoscopy used as a reference standard, high CRP values (>5 mg/L) were associated with high specificity but very low sensitivity. On the other hand, high FC values (>50 mcg/gr) showed opposite characteristics, i.e., high sensitivity but low specificity [24]. Due to SNP polymorphisms, approximately 15% of healthy individuals and up to 20–25% of CD patients do not have a CRP response, which could be one of the explanations for the poor sensitivity reported in many studies [25,26,27]. However, among the therapeutic scenarios in which CRP has been found to be quite reliable is its ability to predict responsiveness to treatment with biological drugs [28]. As proven by observational studies and post-hoc analyses of randomized controlled trials (RCTs), CRP early normalization after anti-TNF-alpha induction or significant reduction (>50%) predicts long-term sustained clinical remission, particularly in CD [7,23].

The Oxford-Travis criteria are a combination of CRP > 45 mg/L with a stool frequency between 3 and 8 per day on the third day of intravenous steroids, which can predict the occurrence of colectomy in acute severe UC (ASUC) during hospitalization [29,30]. FC is the most used biomarker in clinical practice as it correlates well with inflammation in the bowel, including disease activity and is resistant to degradation. However, it has some disadvantages including poorer sensitivity in isolated ileal CD, discomfort when handling liquid stools, the need for standardization (FC cut-offs, FC assay), intra-individual variability (even within a day), and cost [31]. Additionally, FC does not have absolute specificity for IBD as it can be elevated during infections of various etiologies, tumors, drugs, or other clinical conditions that can mimic IBD symptoms (Table 2) [31,32]. FC has been shown to be effective in all major clinical contexts of IBD, including an important role in predicting the effectiveness of biological therapies, similar to CRP [7,23,28]. In fact, FC was able to predict sustained clinical response 1 year after anti-TNF-alpha induction in a prospective study of 63 IBD patients, with 83% sensitivity and 74% specificity using a cut-off of 168 ug/g [33]. Although specific cut-offs capable of predicting response to treatment differ from patient to patient and based on different clinical scenarios, similar results have been observed in post-hoc RCTs and real-life observational studies with biological drugs (i.e., vedolizumab, ustekinumab) and small molecules (i.e., tofacitinib) [7,23,28].

Some novel biomarkers look promising, including blood-based ratios and microRNAs [34]. The neutrophil-to-lymphocyte ratio (NLR) is a readily accessible hematological biomarker that correlates with the response of biological drugs and predicts endoscopic outcomes in both UC and CD [35,36]. On the other hand, distinct panels of miRNAs have been identified in blood and fecal samples of IBD patients, with a future potential role in disease monitoring [34].

### 2.3. Endoscopic Parameters

#### 2.3.1. Ileo-Colonoscopy and Flexible Sigmoidoscopy

Endoscopy (including ileo-colonoscopy (CS) and flexible sigmoidoscopy (SG)) plays a crucial role in the diagnosis and management of IBD since the endoscopic parameters represent the gold standard for evaluating disease activity. It not only helps in the identification of lesions and strictures but also aids in the evaluation of disease activity and response to treatment. Direct visualization of the mucosa and histological confirmation through endoscopy have therapeutic value and are useful in CRC-surveillance. Recently, a promising novel technique called Confocal Laser Endomicroscopy (CLE) may allow in vivo visualization of histological abnormalities and targeted optical biopsies. In particular, recent data supports the idea that increased intestinal permeability as assessed by CLE may predict the relapse of IBD patients, even in the presence of clinical remission, with a future potential role in disease follow-up [37].

Scoring systems have been developed to stratify endoscopic severity, with Simple Endoscopic Score for Crohn’s Disease (SES-CD) and Mayo Endoscopic Sub-score (eMAYO) being the most commonly used in clinical practice [12,38,39]. Studies and meta-analyses have shown that achieving MH is essential in both UC and CD. It has been associated with reduced hospitalization, surgery, bowel damage, corticosteroid use, and risk of clinical flare, as well as a positive impact on work and leisure activities [40,41,42,43,44,45,46,47]. According to a recent meta-analysis of 17 cohort studies that included over 2608 UC patients in clinical remission, those with MES of 0 had a 52% reduced risk of clinical relapse compared to patients with MES of 1. Moreover, individuals who achieved both MES of 0 and histological healing (HH, defined as the absence of neutrophils in the epithelium) had a 63% lower risk of clinical relapse compared to those with histological activity [47].

Despite its effectiveness, endoscopy has some drawbacks. It is an invasive procedure that is not always well accepted by the patient, it is costly, time consuming and it is not without possible complications, although these are rare (<1%) in expert hands [38]. According to a French survey, patients find all monitoring tools offered to them useful (from venipuncture to colonoscopy), but endoscopy is less acceptable when compared to other non-invasive methods such as intestinal ultrasound (IUS), biomarkers, video-capsule endoscopy (VCE), or magnetic resonance imaging [48].

#### 2.3.2. Video-Capsule Endoscopy

Previous studies have shown that 30–70% of CD patients have small bowel involvement and up to 30% of patients diagnosed with CD have only small bowel involvement which increases the likelihood of a more complex disease course [49,50,51,52]. VCE is a non-invasive method that can be used to visualize the small intestinal mucosa accurately. A meta-analysis conducted by Niv revealed that MH detection by capsule is a good predictor of long-term clinical remission [53]. Therefore, the STRIDE-II consensus and recent European Society of Endoscopy (ESGE) guidelines recommend the use of this method for longitudinal monitoring and evaluating the response to medical treatment, provided that specific validated quantitative activity scores (LEWIS score and CECDAI score) are used [7,54]. However, the use of VCE is limited by its cost, restricted availability, risk of capsule retention, and the need for expert training for reporting. Even when a patency capsule is used in individuals judged at high-risk (i.e., those with a history of abdominal surgery or small-bowel obstruction), the rates of capsule retention range from 1.5% to 2.1% [55]. To justify the appropriate use of VCE in long-term monitoring, the combination of VCE and FC may assist [11]. In the future, artificial intelligence (AI) might aid in lower reporting times, reduce interobserver variability, and develop new powered software capable of autonomously classifying inflammatory activity scores [56,57,58].

### 2.4. Radiological Parameters

#### 2.4.1. Magnetic Resonance and Computed Tomography Enterography

Cross-sectional imaging plays a crucial role in the evaluation and monitoring of IBD. It is an effective tool for assessing MH, transmural healing (TH, defined as a bowel wall thickness BWT < 3 mm and no signs of hyperperfusion, edema, ulcers or fat stranding), as well as for monitoring the effects of therapy in CD [11,59,60,61]. In addition to assessing the entire gastrointestinal tract, imaging can also help in the identification of CD complications such as strictures or fistulas, as well as perianal disease. Magnetic resonance enterography (MRE) and computed tomography enterography (CTE) are the most used imaging modalities for CD, although CTE is mostly recommended for emergency settings due to the risk of ionizing radiation exposure but is important to mention that in clinical practice MRE and CTE have comparable diagnostic accuracy [11]. Several scoring systems have been developed to standardize MRE assessment in CD, including the Magnetic Resonance Index of Activity (MaRIA) score, Clermont Index, London score, and Nancy score [60,61]. These scoring systems are effective in correlating with the presence of mucosal inflammation, but they are not easy to use daily. Despite its usefulness, MRE has some limitations, such as limited availability, long image acquisition time, high-cost, the need for fasting, and (sometimes) for bowel preparation [11].

#### 2.4.2. Point-of-Care Ultrasound

Point-of-care ultrasound (POCUS) is gaining significant interest and appeal as a non-invasive, repeatable, low-cost, promising tool for monitoring therapeutic response, and examining patients at the bedside [11,60,62,63]. It is being considered as the future stethoscope [62]. Many studies have demonstrated the clinical utility of IUS for assessing TH [64,65,66], while different activity scores have been developed in both CD and UC. There is now emerging agreement on the most relevant criteria to include and how they should be quantified, although only a few IUS scores for UC have been extensively confirmed and validated [60,61]. The Milan Ultrasound Criteria (MUC) is the simplest score validated in different cohorts for UC [67,68,69,70]. When BWT was assessed under standardized conditions, interrater agreement was good to almost perfect [71,72,73,74]. However, IUS has some limitations, including lower accuracy (compared to MRE) in evaluating the proximal small bowel and rectum, lower interobserver reliability for some sonographic parameters (i.e., color Doppler signal, inflammatory fat detection, bowel wall stratification), and the need for specific training [60,61].

## 3. Discussion

The most recent STRIDE-II international consensus has developed a simple algorithm to meet treatment targets in three main steps. The first step is to make the patient feel well, followed by normalizing biomarkers (serum and fecal levels), and finally achieving MH. Additionally, TH in CD and HH in UC are assessed as measures of the depth of remission achieved, although they are not formal targets. However, it should be noted that this algorithm is a generic scheme with some significant limitations. IUS is not used in this algorithm to track therapeutic response, and some patients with isolated small bowel involvement or those with penetrating/fibrostenotic phenotype may require specific instrumental monitoring [7]. To address these limitations, a simplified evidence-based monitoring algorithm (Table 3) has been presented that can be easily applied in clinical practice and combines all major monitoring modalities, including noninvasive tools such as IUS and VCE.

### 3.1. Clinical Assessment

During active disease, it is recommended to undergo clinical assessment roughly every 3 months (STRIDE-I and II, CALM, ECCO-ESGAR) and approximately every 3–6 months in clinically asymptomatic patients depending upon duration of remission and current therapy (ECCO-ESGAR) [6,7,11,19]. For patients in remission, it is recommended to undergo clinical assessment every 6–12 months (STRIDE-I). The new STRIDE-II recommendations provide some information on the time required to attain treatment goals in UC and CD based on different treatment modalities. However, it should be noted that time points are based on expert opinion and comparisons among different drugs are challenging in the absence of head-to-head controlled trials [7].

### 3.2. Biochemical Assessment

Biochemical assessment is an important tool for determining response to treatment during active disease. According to the ECCO-ESGAR guidelines, FC can be used to determine response to treatment approximately 3–6 months after treatment initiation in active UC and within 3 months in active CD. In IBD patients who have reached clinical and biochemical remission, the interval of monitoring should be between 3–6 months depending upon duration of remission and current therapy [11]. On the other hand, the AGA guidelines recommend 2–4 months of biomarker monitoring for CD patients with symptomatically active disease, and 6–12 months of biomarker monitoring in CD and UC patients in symptomatic remission [20,21].

### 3.3. Endoscopic Evaluation

To monitor therapeutic efficacy, the ECCO-ESGAR guidelines recommend an endoscopic examination 3–6 months after treatment onset in UC and an endoscopic (or transmural) response within 6 months of initiating treatment in CD. A flexible SG is usually sufficient for UC as it involves the mucosa continually from the rectum to the colon [11]. According to the STRIDE-I guidelines, following the initiation of medication, endoscopic disease activity should be reviewed at 6 to 9 months in CD and 3 to 6 months in UC [6]. The AGA guidelines recommend an endoscopic (and/or radiologic) evaluation typically 6–12 months after treatment initiation or adjustment in patients with symptomatically active CD, after resolution of symptoms and normalization of biomarkers [20]. It is important to note that in CD patients who have achieved clinical remission, the AGA guidelines recommend that endoscopic (or radiologic) remission should ideally be confirmed within 3 years [20]. However, in general, endoscopic reassessment of IBD patients should be considered in cases of severe relapse or prompted by consecutive positive biomarkers [76]. The ECCO and British Society of Gastroenterology (BSG) guidelines dictate endoscopic surveillance according to risk stratification as follows:-Patients with high-risk features such as stricture or dysplasia detected within the past 5 years, PSC, extensive colitis with severe active inflammation, or a family history of CRC in a first degree relative less than 50 years should have next surveillance colonoscopy scheduled in 1 year;-Patients with intermediate-risk which includes extensive colitis with mild or moderate active inflammation, post-inflammatory polyps or a family history of CRC in a first degree relative at 50 years and above should have their next surveillance colonoscopy scheduled in 2 to 3 years;-Patients with low-risk such as those with left colitis, pancolitis in endoscopic and histological remission, or Crohn’s colitis affecting less than 50% of the colon, should have their next surveillance colonoscopy scheduled in 5 years [8,9,10,11,75].

In clinical practice, it is imperative to examine MH at the appropriate time, which is neither too early nor too late. A suitable compromise would be to carry out the first colonoscopy roughly one year following the start of therapy, which is also the time period for reassessment inferred from RCTs [11,87]. According to the STRIDE-II consensus and ESGE guidelines, VCE is recommended for evaluating MH in CD patients with small bowel involvement after initiating treatment. However, the surveillance time points in this setting can vary significantly, as reported in a recent systematic review [83]. Some studies used a 3–month interval [88,89] between VCE assessments, while others used a 6–12 month interval [90,91,92]. To minimize the risk of capsule retention, the appropriate use of VCE in long-term monitoring should be prompted by biomarkers positivity. In a prospective study of 43 CD patients in clinical remission, FC (and other fecal biomarkers) was a good predictor of MH assessed by VCE, proving useful in monitoring CD progress. Increased FC levels predicted mucosal inflammation at VCE with an estimated sensitivity of 85%, specificity of 100%, and an AUC of 0.94 using the optimum cutoff of 98 μg/g [84]. These results were confirmed by a recent meta-analysis and systematic review, which found similar pooled results using a cutoff of 100 μg/g (sensitivity of 0.73, specificity of 0.73, and diagnostic odds ratios of 7.89) [85].

### 3.4. Imaging Evaluation

In CD, it is recommended to evaluate TH (using MRE or CTE) ideally after a median of 26 weeks of treatment, according to a recent systematic review that examined the connection between TH and disease-related outcomes in 1835 CD patients [77]. The International Bowel Ultrasound Group suggests that IUS response in IBD patients should initially be assessed at week 14 ± 2 after treatment initiation and between week 26–52, or perform IUS depending on elevated FC, symptoms or clinical suspicion of flare [78,79]. A recent prospective study of UC patients starting biological therapies also demonstrated the usefulness of early IUS (at week 12) with the MUC score in predicting long-term endoscopic response [69]. It is worth noting that IUS response in UC patients treated with small molecules (i.e., tofacitinib) can be identified earlier after treatment begins, and early IUS evaluation may be useful even after 8 weeks [86]. Experts recommend scheduled IUS assessment every 6–12 months in the maintenance phase [62,78,79,80,81,82,86].

## 4. Conclusions

The treat-to-target (T2T) strategy for early and effective treatment is crucial for improving long-term outcomes in IBD. Results of the ongoing REACT-2 trial suggest that T2T based on ulcer healing was no more effective than standard symptom-based management for patients with CD regarding the primary outcome of first complication at 24 months [93]. While STRIDE recommendations provide useful information for tailored treatment management, they have certain limitations, such as the lack of an ultrasound monitoring strategy and being too simple [6,7]. On the other hand, IUS cannot replace endoscopy in the surveillance of UC-associated neoplasia at the moment, but it can reduce the frequency of cumbersome modalities. We presented a schematic evidence-based monitoring algorithm that combines all major monitoring modalities with noninvasive surrogates and can be easily applied in clinical practice. However, more research is needed to develop and validate new diagnostic algorithms for monitoring patients with IBD.

In the future, monitoring tools such as AI, remote monitoring, wearable devices (i.e., fitness bands or watches), and point-of-care technology (i.e., panenteric capsules, Calprosmart, or CRP at home), should be less intrusive, less expensive, and ultimately minimize unnecessary patient hospitalization [94]. Moreover, it is likely that TH will soon become the main therapeutic target in CD.

## Figures and Tables

**Figure 1 jcm-13-01008-f001:**
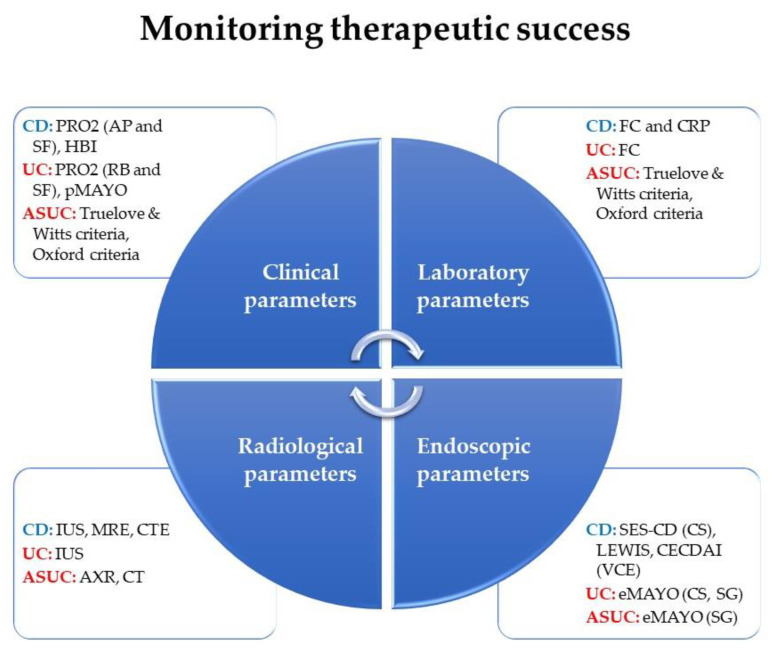
Monitoring therapeutic success scheme according to ECCO-ESGAR diagnostic guidelines. Abbreviations. CD: Crohn’s disease; UC: ulcerative colitis; ASUC: acute severe ulcerative colitis; PRO2: patient-reported outcome; AP: abdominal pain; SF: stool frequency; HBI: Harvey-Bradshaw index; pMAYO: partial Mayo score; eMAYO: endoscopic Mayo score; FC: fecal calprotectin; CRP: C-reactive protein; IUS: intestinal ultrasound; MRE: magnetic resonance enterography; CTE: computed tomography enterography; AXR: abdominal X-ray; CT: computed tomography; SES-CD: simple endoscopic score for Crohn’s disease; CECDAI: capsule endoscopy Crohn’s disease activity index; CS: colonoscopy; SG: sigmoidoscopy.

**Table 1 jcm-13-01008-t001:** Main biomarkers studied in IBD.

Serum Biomarkers	Fecal Biomarkers
CRP, ESR SC Calgranulin C Neopterin Endothelin IL-6, IL-8, IL-17 sTNF-RI, sIL-2R Autoantibodies and Antimicrobial Antibodies (pANCA, ASCA, AMCA, ALCA, ACCA, anti-L, anti-C, anti-CBIR, anti-OMPC, anti-l2) MMP-3, MMP-9 LRG MicroRNAs (MiR-16, MiR-21, MiR-155, Mir-223, Mir-320a) EHI—Endoscopic Healing Index (ANG1, ANG2, CRP, SAA1, IL7, EMMPRIN, MMP1, MMP2, MMP3, MMP9, TGFA, CEACAM1, VCAM1) Blood-based ratios (NLR, PLR, ELR, ENLR)	FC Lactoferrin Eosinophil Cationic Protein M2-Pyruvate Kinase Osteoprotegrin Myeloperoxidase MMP-9 MicroRNAs (MiR-16, MiR-21, MiR-223, MiR-1246) FIT LCN-2/NGAL

Abbreviations. IBD: inflammatory bowel diseases; CRP: C-reactive protein; ESR: erythrocyte sedimentation rate; SC: serum calprotectin; sTNF-RI: human soluble tumor necrosis factor receptor I; sIL-2R: soluble interleukin-2 receptor; pANCA: perinuclear antineutrophil cytoplasmic; ASCA: anti-saccharomyces cerevisiae antibodies; AMCA: anti-mannobioside carbohydrate antibody; ALCA: anti-chitobioside carbohydrate antibody; ACCA: anti-chitobioside carbohydrate antibody; anti-L: anti-laminarin; anti-C: anti-chitin; anti-CBIR: anti-flagellin CBir1; anti-OMPC: antibodies to the outer-membrane porin C of escherichia coli; anti-l2: antibodies against a pseudomonas fluorescens-associated sequence I2; MMP: matrix metalloproteinase; LRG: serum leucine-rich glycoprotein; ANG: angiopoietin; SAA1: serum amyloid A1; EMMPRIN: extracellular matrix metalloproteinase inducer; TGFA: transforming growth factor alpha; CEACAM1: carcinoembryonic antigen related cell adhesion molecule 1; VCAM1: vascular cell adhesion molecule 1; NLR: neutrophil-to-lymphocyte ratio; PLR: platelet-to-lymphocyte ratio; ELR: eosinophil-to-lymphocyte ratio; ENLR: eosinophil*neutrophil-to-lymphocyte ratio; FC: fecal calprotectin; FIT: fecal immunochemical test; LCN-2/NGAL: lipocalin-2/neutrophil gelatinase-associated lipocalin.

**Table 2 jcm-13-01008-t002:** Examples of non-IBD causes of increased FC.

Non-IBD Causes of Increased FC	
Infections	Other GI diseases
Giardia lamblia Bacterial gastroenteritis (Salmonella, Campylobacter, CDI) Viral gastroenteritis (Rotavirus, Norovirus, SARS-CoV2) Helicobacter pylori gastritis Respiratory infections	Cystic fibrosis Coeliac disease (untreated) Diverticular disease SCAD Appendicitis Protein losing enteropathy Colorectal adenoma Juvenile polyp Autoimmune enteropathy Microscopic colitis Necrotising enterocolitis GVHD Liver cirrhosis Young age (infants < 4 years) Family history of IBD (1st degree relative)
Malignancies	
CRC Gastric carcinoma Intestinal lymphoma	
Drugs	
NSAID PPI Sartans Levodopa	

Abbreviations. IBD: inflammatory bowel diseases; FC: fecal calprotectin; CDI: clostridioides difficile infection; SARS-CoV2: severe acute respiratory syndrome coronavirus 2; CRC: colorectal cancer; NSAID: non-steroidal anti-inflammatory drugs; PPI: proton pump inhibitors; SCAD: segmental colitis associated with diverticulosis; GVHD: graft-versus-host disease.

**Table 3 jcm-13-01008-t003:** Potential timing for monitoring IBD patients in clinical practice.

Crohn’s Disease		
	Active Phase	Maintenance Phase
Clinical assessment (PRO2, HBI)	3 M [6,7,11,12,19]	Every 3–6–12 M [6,11,12]
Biochemical assessment (CRP, FC)	2–4 M after therapy [7,11,12,20]	Every 3–6–12 M [11,12,20]
Endoscopic evaluation (SES-CD)	6–9 M after therapy (within 1 Y) [6,11,12,20]	Within 3 Y [20] Based on surveillance recommendations [8,10,75] Prompted by clinical symptoms or biomarkers positivity (FC) [11,12,76]
Imaging evaluation (IUS/MRE)	3–6–12 M after therapy [11,12,20,77,78]	Within 3 Y [20] Every 6–12 M [62,78,79,80,81,82] Prompted by clinical symptoms or biomarkers positivity (FC) [78]
VCE (LEWIS, CECDAI)	3–6–12 M after therapy [83]	Prompted by biomarkers positivity (FC) [84,85]
Ulcerative colitis		
Clinical assessment (PRO2, pMAYO)	3 M [6,7,11,12]	Every 3–6–12 M [6,11,12]
Biochemical assessment (CRP, FC)	3–6 M after therapy [7,11,12]	Every 3–6–12 M [11,12,21]
Endoscopic evaluation (eMAYO)	3–6 M after therapy (within 1 Y) [6,11,12]	Based on surveillance recommendations [8,10,75] Prompted by clinical symptoms or biomarkers positivity (FC) [11,12,76]
Imaging evaluation (IUS)	2–3–6 M after therapy [69,78,86]	Every 6–12 M [62,78,79,80,82] Prompted by clinical symptoms or biomarkers positivity (FC) [78]

Abbreviations. IBD: inflammatory bowel diseases; M: months, Y: years; PRO2: patient-reported outcome; HBI: Harvey–Bradshaw index; CRP: C-reactive protein; FC: fecal calprotectin; SES-CD: simple endoscopic score for Crohn’s disease; IUS: intestinal ultrasound; MRE: magnetic resonance enterography; VCE: video-capsule endoscopy; CECDAI: capsule endoscopy Crohn’s disease activity index; pMAYO: partial Mayo score; eMAYO: endoscopic Mayo score.

## Data Availability

Not applicable.

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
