# Peer review of "Current Approaches for Monitoring of Patients with Inflammatory Bowel Diseases: A Narrative Review"

_jcm, 2024, doi:10.3390/jcm13041008_

Round 1

Reviewer 1 Report

Comments and Suggestions for Authors

Minor concerns

1.       Page 2, Figure 1, the text within the figure is not completely visible, please modify accordingly.

2.       Page 3, Table 1. Please avoid using ellipsis dots within the table.

Major concerns

3.       In your review you do not include many recent published articles, with only 14 articles dating from 2023 and 7 articles dating from 2022 from 83 citations. Please, include more recent articles.

4.       You did not mention or discuss the potential use of Confocal Laser Endomicroscopy (CLE) in the follow-up of patients with IBD. Recent data supports that increased intestinal permeability as assessed by CLE could predict relapse in patients with IBD, even in the presence of clinical remission (Chiriac S, et al. Impaired Intestinal Permeability Assessed by Confocal Laser Endomicroscopy-A New Potential Therapeutic Target in Inflammatory Bowel Disease. Diagnostics (Basel). 2023 Mar 24;13(7):1230. doi: 10.3390/diagnostics13071230. PMID: 37046447; PMCID: PMC10093200.). Please, comment on this.

Comments on the Quality of English Language

The English level is acceptable, minor changes are required.

Author Response

Minor concerns

  1. Page 2, Figure 1, the text within the figure is not completely visible, please modify accordingly.

R: As suggested, the Figure 1 (page 2, line 54) has been edited ed improved.

  1. Page 3, Table 1. Please avoid using ellipsis dots within the table.

R: We thank the reviewer for this advice. The table's ellipsis dots (page 3, line 108) have now been removed.

Major concerns

  1. In your review you do not include many recent published articles, with only 14 articles dating from 2023 and 7 articles dating from 2022 from 83 citations. Please, include more recent articles.

R: We thank the reviewer for this advice. We added the following, more recent, articles:

10) Gordon H, Biancone L, Fiorino G, et al. ECCO Guidelines on Inflammatory Bowel Disease and Malignancies. J Crohns Colitis. 2023.

37) Chiriac S, Sfarti CV, Minea H, et al. Impaired Intestinal Permeability Assessed by Confocal Laser Endomicroscopy-A New Potential Therapeutic Target in Inflammatory Bowel Disease. Diagnostics (Basel). 2023.

58) Diaconu C, State M, Birligea M, et al. The Role of Artificial Intelligence in Monitoring Inflammatory Bowel Disease-The Future Is Now. Diagnostics (Basel). 2023.

63) Dolinger MT, Kellar A. Point-of-Care Intestinal Ultrasound in Pediatric Inflammatory Bowel Disease. Curr Gastroenterol Rep. 2023.

61) Hameed M, Taylor SA. Small bowel imaging in inflammatory bowel disease: updates for 2023. Expert Rev Gastroenterol Hepatol. 2023.

35) Magalhaes D, Peyrin-Biroulet L, Estevinho MM, Danese S, Magro F. Pursuing neutrophils: systematic scoping review on blood-based biomarkers as predictors of treatment outcomes in inflammatory bowel disease. Therap Adv Gastroenterol. 2023.

34) Alghoul Z, Yang C, Merlin D. The Current Status of Molecular Biomarkers for Inflammatory Bowel Disease. Biomedicines. 2022.

36) Crispino F, Grova M, Maida M, et al. Blood-based prognostic biomarkers in Crohn's Disease patients on biologics: a promising tool to predict endoscopic outcomes. Expert Opin Biol Ther. 2021.

64) Castiglione F, Imperatore N, Testa A, et al. One-year clinical outcomes with biologics in Crohn's disease: transmural healing compared with mucosal or no healing. Aliment Pharmacol Ther. 2019. 

65) Rispo A, Imperatore N, Testa A, et al. Combined Endoscopic/Sonographic-based Risk Matrix Model for Predicting One-year Risk of Surgery: A Prospective Observational Study of a Tertiary Centre Severe/Refractory Crohn's Disease Cohort. J Crohns Colitis. 2018.

66) Castiglione F, Mainenti P, Testa A, et al. Cross-sectional evaluation of transmural healing in patients with Crohn's disease on maintenance treatment with anti-TNF alpha agents. Dig Liver Dis. 2017.

  1. You did not mention or discuss the potential use of Confocal Laser Endomicroscopy (CLE) in the follow-up of patients with IBD. Recent data supports that increased intestinal permeability as assessed by CLE could predict relapse in patients with IBD, even in the presence of clinical remission (Chiriac S, et al. Impaired Intestinal Permeability Assessed by Confocal Laser Endomicroscopy-A New Potential Therapeutic Target in Inflammatory Bowel Disease. Diagnostics (Basel). 2023 Mar 24;13(7):1230. doi: 10.3390/diagnostics13071230. PMID: 37046447; PMCID: PMC10093200.). Please, comment on this.

R: We thank the reviewer for this valuable advice. We have updated the endoscopic paragraph mentioning the technique and including a reference for CLE technique (page 5, lines 142-147).

Reviewer 2 Report

Comments and Suggestions for Authors

Dear Colleagues,

I have completed a thorough review of the review titled "Current approaches for monitoring of patients with inflammatory bowel diseases: a narrative review" submitted to Journal of Clinical Medicine. The authors have provided a comprehensive description of the current approaches for monitoring of patients with IBD. Authors effectively summarized these approaches through the use of figures and tables. The review is well-written and provides sufficient details. However, I would like to draw attention to an area that could benefit from further refinement. While the authors have covered laboratory parameters extensively, specifically focusing on CRP and FC, there is a notable absence of commentary on potential new serum and fecal biomarkers. I recommend that the authors address this gap by incorporating a discussion on the potential of at least one additional serum biomarker and fecal marker, thereby enriching the manuscript's content.

Additionally, I observed a formatting issue in Figure 1, where the text within the four boxes is partially obscured by white squares. I recommend rectifying this matter to ensure clarity.

I believe these suggested revisions will strengthen the review and enhance its scientific rigor.

Author Response

-While the authors have covered laboratory parameters extensively, specifically focusing on CRP and FC, there is a notable absence of commentary on potential new serum and fecal biomarkers. I recommend that the authors address this gap by incorporating a discussion on the potential of at least one additional serum biomarker and fecal marker, thereby enriching the manuscript's content.

R: We agree with this comment. A brief discussion of two potential new biomarkers (serum and fecal) has been now included (page 4, lines 127-132).

-Additionally, I observed a formatting issue in Figure 1, where the text within the four boxes is partially obscured by white squares. I recommend rectifying this matter to ensure clarity. I believe these suggested revisions will strengthen the review and enhance its scientific rigor.

R: We agree with this comment. As suggested, the Figure 1 (page 2, line 54) has been edited ed improved.